# Giant room temperature electrocaloric effect in a layered hybrid perovskite ferroelectric: [(CH$_3$)$_2$CHCH$_2$NH$_3$]$_2$PbCl$_4$

Xitao Liu [1,2✉], Zhenyue Wu[1,2,3], Tong Guan[4], Haidong Jiang[4], Peiqing Long[1,2], Xiaoqi Li[1,2], Chengmin Ji[1,2], Shuang Chen [4✉], Zhihua Sun [1,2,3] & Junhua Luo [1,3✉]

Electrocaloric effect driven by electric fields displays great potential in realizing highly efficient solid-state refrigeration. Nevertheless, most known electrocaloric materials exhibit relatively poor cooling performance near room temperature, which hinders their further applications. The emerging family of hybrid perovskite ferroelectrics, which exhibits superior structural diversity, large heat exchange and broad property tenability, offers an ideal platform. Herein, we report an exceptionally large electrocaloric effect near room temperature in a designed hybrid perovskite ferroelectric [(CH$_3$)$_2$CHCH$_2$NH$_3$]$_2$PbCl$_4$, which exhibits a sharp first-order phase transition at 302 K, superior spontaneous polarization (>4.8 $\mu$C/cm$^2$) and relatively small coercive field (<15 kV/cm). Strikingly, a large isothermal entropy change $\Delta S$ of 25.64 J/kg/K and adiabatic temperature change $\Delta T$ of 11.06 K under a small electric field $\Delta E$ of 29.7 kV/cm at room temperature are achieved, with giant electrocaloric strengths of isothermal $\Delta S/\Delta E$ of 0.86 J·cm/kg/K/kV and adiabatic $\Delta T/\Delta E$ of 370 mK·cm/kV, which is larger than those of traditional ferroelectrics. This work presents a general approach to the design of hybrid perovskite ferroelectrics, as well as provides a family of candidate materials with potentially prominent electrocaloric performance for room temperature solid-state refrigeration.

[1] State Key Laboratory of Structural Chemistry, Fujian Institute of Research on the Structure of Matter, Chinese Academy of Sciences, Fuzhou, China. [2] Fujian Science & Technology Innovation Laboratory for Optoelectronic Information of China, Fuzhou, Fujian, China. [3] University of Chinese Academy of Sciences, Chinese Academy of Sciences, Beijing, China. [4] Kuang Yaming Honors School and Institute for Brain Sciences, Nanjing University, Nanjing, Jiangsu, China. ✉email: xtliu@fjirsm.ac.cn; chenshuang@nju.edu.cn; jhluo@fjirsm.ac.cn

Refrigeration is playing a more and more important role in modern society, such as for air conditioning, food storage, and industrial manufacture[1,2]. Present-day refrigeration technology primarily relies on conventional mechanical vapor-compression of gases, which has been proven to grow serious environmental concerns. As a consequence, caloric materials that exhibit solid-state ferroic phase transitions under external stimulus (magnetic, electric, or mechanic) have attracted significant attention[3–8]. Among them, electrocaloric (EC) materials that are capable of heat exchange driven by electric fields have achieved a particularly important status due to their high efficiency, natural integration, and quiet operation[9–11]. The prototype EC materials are ferroelectrics, which exhibit a reversible spontaneous polarization under external electric fields[12]. Since 1930s[13], great efforts have been made for searching appropriate ferroelectrics to realize high efficient solid-state EC refrigeration. So far, a variety of inorganic perovskite and organic polymer ferroelectrics have been explored, such as $KH_2PO_4$[14], $BaTiO_3$[15], $Pb(Zr,Ti)O_3$[9], triglycine sulfate (TGS)[16], and poly(vinylidene fluoride) (PVDF)[10]. Nevertheless, most of them suffer from poor cooling performance, large driving electric fields, and operational temperature far away from room temperature, which becomes the potential bottleneck to their practical applications in cooling devices. In this context, it is highly imperative to develop ferroelectric materials which are capable of generating a giant EC effect near room temperature for high-performance solid-state refrigeration.

Inspired by the breakthrough development of $CH_3NH_3PbI_3$, the family of organic–inorganic hybrid perovskites has emerged as one of the most active fields of condensed matter, which exhibits widespread potentials in electric, magnetic, luminescent, and photovoltaic devices[17–19]. Structurally, the integration of organic and inorganic components endows hybrid perovskites with exceptional structural flexibility and chemical diversity[20,21], which provides an ideal platform for structural design and material exploration. Meanwhile, the versatile structures of hybrid perovskites enable a fine-tuning of optoelectronic properties through compositional engineering[22,23]. More importantly, the dynamic organic cations residing between the inorganic perovskite frameworks offer a large degree of molecular motion freedom, creating a driving force to generate multiple order–disorder phase transitions associated with large entropy changes[24–28]. Nowadays, large entropy changes associated with ferroic phase transitions have been achieved in numerous hybrid perovskites, such as metal-free perovskite ferroelectric $(MDABCO)(NH_4)I_3$[29], double perovskite ferroelectric $(Me_3NOH)_2KFe(CN)_6$[30], hybrid perovskite ferroic [TPrA][Mn(dca)_3][31], layered perovskite ferroelectric $(C_6H_{11}NH_3)_2PbBr_4$[32], triggered by the dynamic ordering of organic cations under the external stimulus. However, an aspect that has not yet been explored in hybrid perovskites for room temperature EC materials. Taking into consideration of the superior structural diversity, large heat exchange, and broad property tenability, it is expected that hybrid perovskites could offer great opportunities to achieve high-performance room temperature EC refrigeration.

Herein, we report a giant EC effect near room temperature in a designed layered hybrid perovskite ferroelectric $(iBA)_2PbCl_4$ (1), where iBA is iso-butylammonium, $(CH_3)_2CHCH_2NH_3$. A large entropy change $\Delta S > 30.26$ J/kg/K accompanying a sharp first-order phase transition was observed near room temperature (302 K), indicating a broad range of practical cooling applications of 1. Furthermore, 1 exhibits an exceptionally large EC effect with giant EC strengths of isothermal $\Delta S/\Delta E$ of 0.86 J cm/kg/K/kV and adiabatic $\Delta T/\Delta E$ of 370 mK cm/kV even under a small electric field $\Delta E$ of 29.7 kV/cm. To the best of our knowledge, this is the first case of hybrid perovskite ferroelectrics with giant room temperature EC performance. This finding opens up great opportunities for the design of hybrid perovskite ferroelectrics

and advances the solid-state refrigeration technologies based on hybrid perovskites.

## Results

Bulk transparent single crystals of 1 (Supplementary Fig. 1) were grown from a saturated solution of concentrated hydrochloric acid containing stoichiometric amounts of $Pb(COOH)_2 \cdot 3H_2O$ and isobutylamine by the temperature lowing method. The obtained single crystals were transparent colorless plate like sheets with dimensions of millimeter-scale. Observation under a polarizing microscope confirmed that the grown single crystals are in the single-domain state. The phase purity of grown crystals was identified by powder X-ray diffraction (XRD, Supplementary Fig. 2), which shows good environmental and electrical stability (Supplementary Figs. 3, 4). As illustrated in Fig. 1, single-crystal XRD analyses (Supplementary Data 1–3) indicate that 1 features a typical layered hybrid perovskite structure with a formula of $(iBA)_2PbCl_4$. The crystal structure contains a single inorganic perovskite layer sandwiched between interlayer organic iBA cations. Interestingly, the inorganic perovskite layer with corner-sharing $PbCl_6$ octahedra creates an open two-dimensional confined space for the interlayer organic iBA cations, offering infinite possibilities to generate multiple phase transitions that respond to external stimuli. At a low temperature of 285 K, 1 crystallizes in a polar space group of Pm (Fig. 1a and Supplementary Fig. 5, Supplementary Table 1). The inorganic $PbCl_6$ octahedra feature a highly distorted configuration, as well as the organic iBA moieties closely pack in an ac plane with highly ordered states. The strong coupling of purposeful rotations of $PbCl_6$ octahedra and cation ordering of iBA cations makes the positive and negative charge centers shift oppositely, which creates a parallel alignment of adjacent dipoles and gives rise to a macroscopic overall non-zero spontaneous polarization in 1. As expected, with temperature increasing, crystal 1 passes through a middle phase with a centrosymmetric space group of Pnma (318 K, Fig. 1b), and finally transforms to a higher symmetric space group of Cmca (343 K, Fig. 1c), belonging to the 88 potential ferroelectric phase transitions (Supplementary Fig. 6)[33,34]. The most evident change at high temperature is that the iBA cations turn into highly disordered states, as well as the distorted inorganic $PbCl_6$ octahedra transform into highly symmetric configurations. In contrast to traditional inorganic perovskite ferroelectrics[9,15,35–37], which traditionally display purely displacive phase transitions accompanied with small entropy changes, the high degree of ordering and disordering of dipoles in hybrid perovskite structure of 1 will result in multiple ferroic orders associated with large isothermal entropy changes. As a consequence, it is reasonable to expect an exceptionally large EC effect in 1.

Since ferroelectric materials generally exhibit the largest EC response close to the ferroelectric–paraelectric (FE–PE) phase transitions[3,38], the phase transition behaviors of 1 were systematically investigated. The preliminary differential scanning calorimetry (DSC) measurements present two sharp thermal peaks located at 307/297 K ($T_1$) and 335/329 K ($T_2$), revealing two reversible phase transitions in 1 upon heating and cooling (Fig. 2a and Supplementary Figs. 7, 8). Meanwhile, temperature-dependent dielectric anomalies (Fig. 2b and Supplementary Fig. 9, 10), in which a step-like anomaly at $T_1$ and a large peak-like anomaly at $T_2$ also probe the two dramatic phase transitions. It is worth mentioning that only the heat exchanges around $T_1$ are focused on since the FE-PE phase transition of 1 center at $T_1 \approx 302$ K ($T_C$, determined by the following physical measurements). The wide thermal hysteresis of 10 K combined with the sharp anomalous peaks at $T_1$ reveals that 1 undergoes a sharp first-order phase transition at $T_C$, which has been considered the most appropriate mechanism for solid-state refrigeration[3,38].

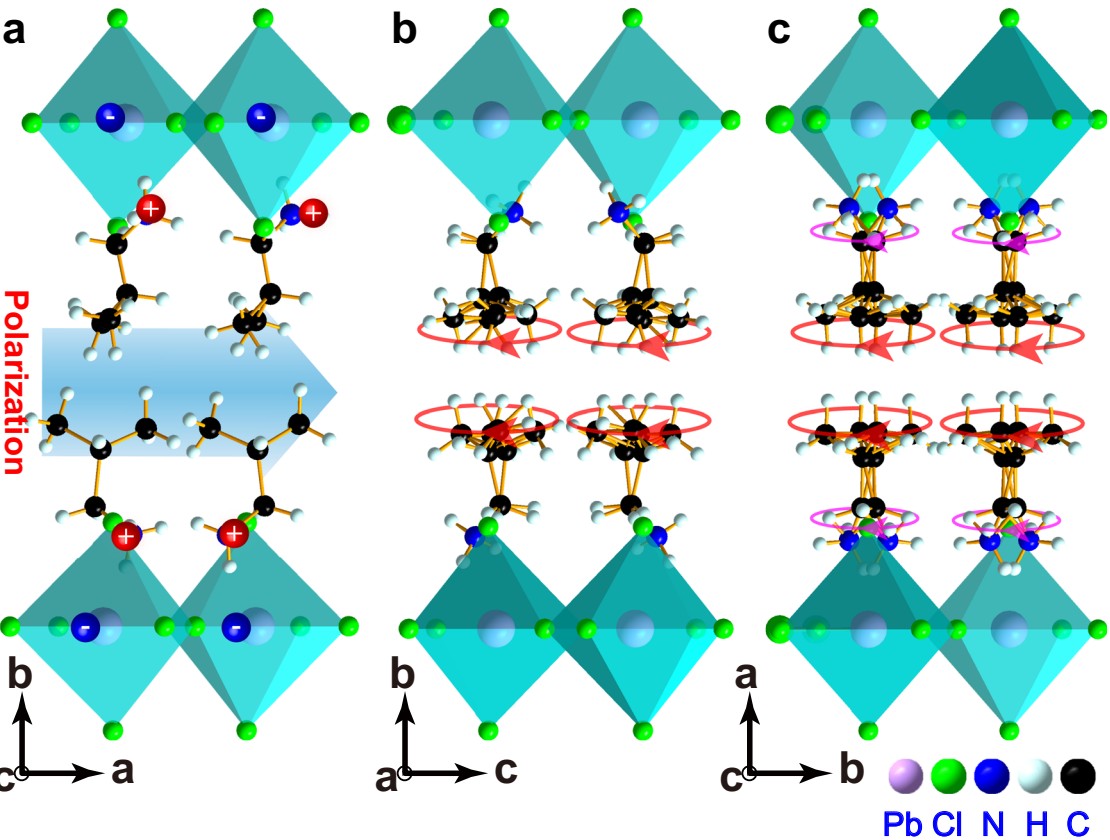

**Fig. 1 Illustration of phase transitions of 1. a** Monoclinic phase at 285 K. **b** Orthorhombic phase at 318 K. **c** Orthorhombic phase at 343 K.

Notably, the $T_C$ (302 K) locates at the room temperature range, suggesting a broad range of practical cooling applications of **1**. Integration of $(dQ/dT)/T$ around $T_C$ during the heating and cooling recycles yields an entropy change ($\Delta S$) of 30.26 J/(kg K) (Fig. 1a, bottom panel) with a latent heat ($Q$) of 9140 J/kg. These values are far larger than those of classic inorganic ferroelectrics, such as $BaTiO_3$[15], $Pb(Ti, Zr)O_3$[9], and $KH_2PO_4$[14], and comparable to those of organic ferroelectric polymers, such as PVDF[10]. Consequently, manipulating the large latent heat derived from the sharp first-order phase transitions around $T_C$ by external electric fields will result in a giant room temperature EC effect in **1**.

It is known that the EC effect is the converse of the pyroelectric effect in which changes of temperature modify electrical polarization. As shown in Fig. 3a, pyroelectric measurements at zero-bias demonstrate that as the temperature increasing, the polarization ($P$) sharply decreases from 4.9 to 0 μC/cm² at 302 K, resulting in a large polarization change with ultrahigh pyroelectric coefficients ($\partial P/\partial T$) up to $252 \times 10^{-4}$ C/m²/K. Furthermore, the polarization change as temperature increases by **1** was further confirmed by temperature-dependent second harmonic generation (SHG) response. It is shown that **1** exhibits a response approximate to that of $KH_2PO_4$ below $T_1$ (Fig. 3b), consistent with its polar characteristic. Meanwhile, as the temperature increasing over $T_1$, the SHG signal of **1** gradually decrease to almost zero, suggesting a centrosymmetric feature. Considering the large polarization change and pyroelectricity near $T_C$, it is expected that **1** features a remarkable EC performance near room temperature.

A large EC effect requires a large entropy change associated with the polarization change, and the ferroelectric material must be capable of generating large polarization changes. Temperature dependence polarization–electric field ($P$–$E$) hysteresis loops at 50 Hz were performed to further investigate the ferroelectric and EC performance in **1**. Figure 4a shows the $P$–$E$ measurements at

temperature from 293 to 306 K, across the $T_C$ of 302 K. It is worth noticing that typical rectangular $P$–$E$ ferroelectric loops are observed in a broad temperature range. At 293 K, the spontaneous polarization is derived to be 4.8 μC/cm² with a coercive electric field of 15 kV/cm (Supplementary Fig. 11), which is much lower than those of inorganic perovskite ferroelectrics and organic ferroelectric polymers, such as PVDF (>100 kV/cm)[39]. Considering the possibility of antiferroelectricity of the intermediate phase, the $P$–$E$ loops under a much higher electric field have been carried out. Unfortunately, it is failure to obtain the double hysteresis loops until the breakdown of the crystal samples (>60 kV/cm). As shown in Fig. 4b and Supplementary Fig. 12, as temperature increases, $P$ of **1** at different electric fields monotonously decreases with a peak drop at 302 K, which is consistent with DSC and pyroelectric measurements. It is worth noticing that a fast polarization change over 70% is observed within only 3 K, implying a significant $\Delta S$ associated to the sharp first-order phase transition at $T_C$. Accordingly, the EC effect in **1** is evaluated by an indirect method based on Maxwell's relations[40]. As shown in Fig. 4c, the crystal features a maximum isothermal $\Delta S$ of 25.64 J/(kg K)) and adiabatic $\Delta T$ of 11.06 K at $\Delta E$ of 29.7 kV/cm with temperature increasing. The corresponding latent heat ($Q$) with a maximum of 7743 J/kg at $\Delta E$ of 29.7 kV/cm, is in good agreement with the heat flow integration results (Fig. 1a). That is to say that under electrical breakdown strength, layered hybrid perovskite ferroelectrics could realize large $\Delta S$ and $\Delta T$ changes by applying relatively small $\Delta E$. These measured giant isothermal entropy and adiabatic temperature spans convert **1** into very attractive EC materials.

## Discussion

In comparison between **1** and other known promising EC materials in terms of cooling performance (Table 1 and Fig. 4e), it can be

noticed that the operating temperature of **1** is near room temperature (302 K), which is crucial for practical cooling applications. On the other hand, single crystals of **1** exhibit a high value of $\Delta S$ (17.44 J/(kg K)) and $\Delta T$ (6.58 K) even applied a small $\Delta E$ of 15.2 kV/cm. In contrast with other known promising EC materials, the EC strengths, which is defined as isothermal $\Delta S/\Delta E$ and adiabatic $\Delta T/\Delta E$, for **1** can achieve as high as 1.15 J cm/kg/K/kV and

430 mK cm/kV at $\Delta E$ of 15.2 kV/cm as well as 0.86 J cm/kg/K/kV and 370 mK cm/kV at $\Delta E$ of 29.7 kV/cm, which are more than two times the existing record values obtained in BaTiO$_3$ single crystals and almost one order of magnitude larger than that of traditional ferroelectric materials[15], representing a record among all the room temperature EC materials reported to date (Table 1)[10,36,37,41–43]. In other way, theoretical approach represents an important advance to quantify the EC effect by using ab initio calculations[11,44–46]. Theoretical investigation based on first-principle effective Hamiltonian approach[11] (Details have been added in Supplementary Methods), which represents a very attractive and powerful method for the quantitative prediction of EC properties. As shown in Fig. 5a and b, our computational isothermal entropy changes and adiabatic temperature changes based on the Landua–Ginzburg theory indicate that the EC performance of (iBA)$_2$PbCl$_4$ is almost one order magnitude larger than those of the traditional ferroelectrics, with an entropy change $\Delta S$ of about 22.5 J/kg/K and an adiabatic temperature change $\Delta T$ of about 7.9 K at room temperature under the applied electric field of 23.2 kV/cm, which is in consistent with the results evaluated by indirect method. The underlying mechanism of the prominent EC performance may be attributed to the large polarization change and relatively small driving electric field required to generate a sharp phase transition. These results indicate that the family of hybrid perovskite ferroelectric materials as the promising candidates for room temperature EC refrigeration.

In summary, we report a giant EC effect near room temperature in a developed layered hybrid perovskite ferroelectric. It exhibits a sharp first-order phase transition at 302 K, superior spontaneous polarization (>4.8 µC/cm$^2$), and a relatively small coercive field (15 kV/cm). More importantly, benefiting from the large entropy changes associated with the sharp first-order phase transition, **1** exhibits exceptionally large EC effect at room temperature with giant EC strengths of isothermal $\Delta S/\Delta E$ of 0.86 J cm/kg/K/kV and adiabatic $\Delta T/\Delta E$ of 370 mK cm/kV even under a small $\Delta E$ of 29.7 kV/cm. Considering structural variability and tenability of hybrid perovskites, an exceptionally large EC effect from hybrid perovskites presents a very promising approach for practical room temperature refrigeration. In particular, it is expected that a fine substitution of iBA components by cations with larger dipole moment and more flexibility will generate much higher ferroelectricity and EC performance. These findings bring exciting prospects in the field of solid-state refrigeration based on ferroic hybrid perovskites.

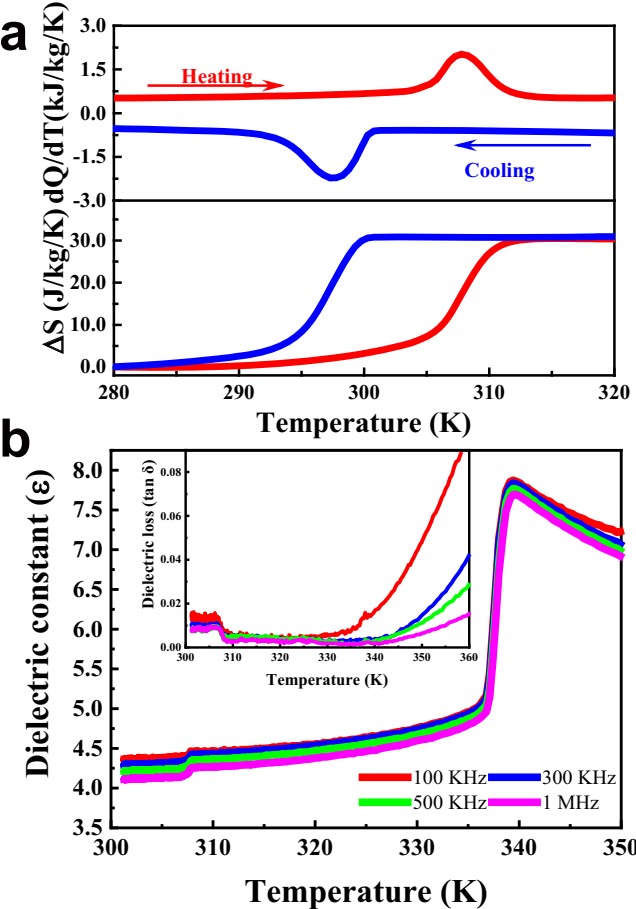

**Fig. 2 Phase transitions in 1. a** Top panel, heat flow $dQ/dT$ on heating (red) and cooling (blue) across the FE–PE transition. Bottom panel, the resulting temperature dependence entropy change $\Delta S$. **b** Temperature-dependent relative permittivities. Inset is the temperature-dependent dielectric loss (tan $\delta$).

## Methods

**Synthesis and crystal growth.** All commercially available reagents were analytical (AR) grade and purchased without further purification. Single-phase is polycrystalline [(CH$_3$)$_2$CHCH$_2$NH$_3$]$_2$PbCl$_4$ was synthesized from an aqueous solution comprising of *iso*-butylamine, Pb(CH$_3$COO)$_2$·3H$_2$O, in a ratio of 1:1 in concentrated hydrochloric acid at ambient temperature. First, 1.14 g Pb(CH$_3$COO)$_2$·3H$_2$O (3 mmol) were slowly

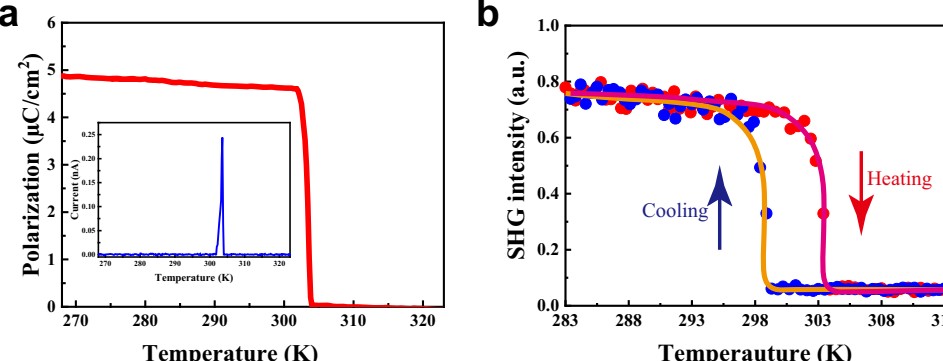

**Fig. 3 Physical properties related to phase transition and symmetry breaking. a** Temperature-dependent spontaneous polarization obtained from the integration of pyroelectric current. Inset: pyroelectric current measured with temperature increasing. **b** Temperature-dependent SHG response.

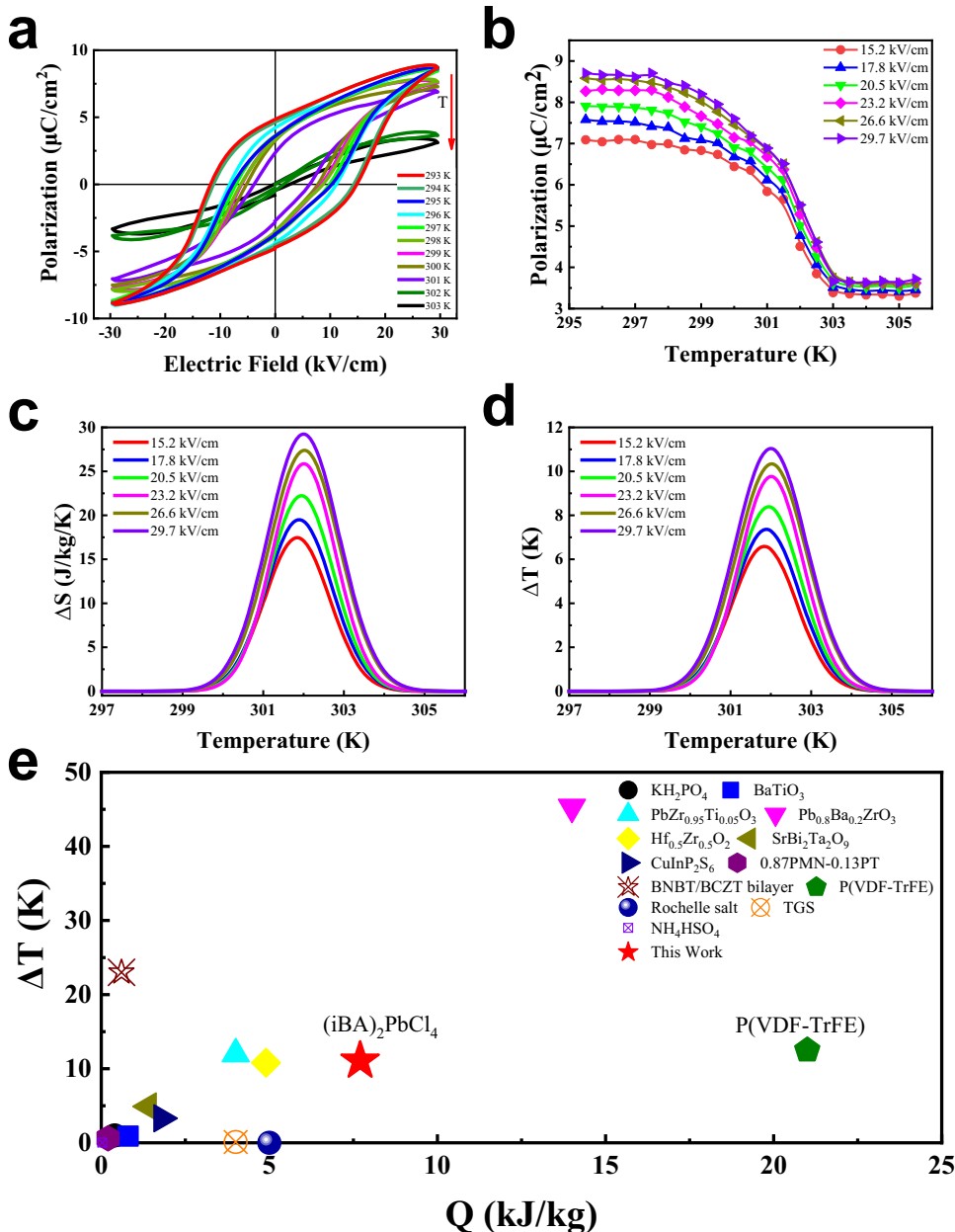

**Fig. 4 Ferroelectric and EC performance. a** Polarization versus electric field characteristics measured at different temperatures. **b** Temperature-dependent polarization at different electric fields, extracted from (**a**). **c** Temperature-dependent adiabatic temperature change ΔT at different ΔE. **d** Temperature-dependent isothermal entropy changes ΔS at different ΔE. **e** Comparison in terms of cooling performance between **1** and a number of known EC materials.

added into 20 mL hydrochloric acid (40%). Then, *iso*-butylamine cation (1.1 g, 15 mmol) was carefully added under stirring and the resultant was obtained after a few minutes. Bulk single crystals were grown from a saturated solution by the temperature lowering method with a cooling rate of 0.5 °C/day. The saturated solution of **1** was prepared at about 50 °C and then kept 4–6 °C above its preliminary saturated temperature for 10 h, which can ensure the dissolution of all ingredients. Then, the saturated solution was slowly cooling with a temperature rate of 0.2 K/day. Finally, the pristine crystals with dimensions up to $15 \times 5 \times 0.5$ mm$^3$ (Supplementary Fig. 1). Before the physical properties measurements, the grown single crystals were dried in an $N_2$ glovebox for 24 h and then annealed at 70 °C for 2 h.

**Crystal structure determination and refinements**. Variable-temperature crystal structure data at 285, 318 and 343 K was collected on a Bruker D8 single-crystal X-ray diffractometer (Mo Kα, $\lambda = 0.71073$ Å). The structures were solved with the ShelXS structure solution program by direct methods and refined with the ShelXL software package using Olex2 software. The refinements were undertaken with anisotropic atomic displacements for all non-hydrogen atoms. The positions of hydrogen atoms of *n*-BA cations were calculated geometrically. The details crystal data and refinement information at 285, 318, and 343 K are illustrated in Table S1.

**Calculation of ΔS and N**. DSC measurements were carried out on a high precision DSC instrument (NETZSCH DSC 200 F3) with a scan rate of 10 K/min in a nitrogen atmosphere. The sapphire sample was used as the standard to calculate the specific heat values. According to Boltzmann equation:

$$\Delta S = R\ln(N) \tag{1}$$

where ΔS is the entropy change; R is the gas constant (8.314 J/(mol K)); N is the number of possible orientations for the disordered system.

**SHG activities measurements**. SHG measurements were performed on polycrystals by the Kurt–Perry method using an Nd:YAG laser ($\lambda = 1064$ nm, 5 ns pulse duration, ~1.6 MW peak power, 10 Hz repetition rate). Temperature-dependent SHG responses were performed using the ground powders with a particle size range of 48–120 μm from 283 to 313 K.

**Electrical measurements and calculation of EC effect**. Electrical measurements were conducted on pieces of crystal wafers cutting perpendicular to the [100] crystallographic direction, which were covered with silver conducting paste on the

**Table 1 Selected electrocaloric effects at phase transitions near and away from room temperature.**

| EC material | $T_C$ (K) | $\Delta S$ (J/kg/K) | $Q$ (kJ/kg) | $\Delta T$ (K) | $\Delta E$ (kV/cm) | $\Delta S/\Delta E$ (J cm/kg/K/kV) | $\Delta T/\Delta E$ (mK cm/kV) | Ref. |
|---|---|---|---|---|---|---|---|---|
| $KH_2PO_4$ | 123 | 3.5 | 0.4 | 1.0 | 10 | 0.35 | 100 | 14 |
| $BaTiO_3$ | 397 | 2.1 | 0.8 | 0.87 | 4 | 0.525 | 220 | 15 |
| $PbZr_{0.95}Ti_{0.05}O_3$ | 499 | 8.0 | 4 | 12 | 480 | 0.017 | 25 | 9 |
| $Pb_{0.8}Ba_{0.2}ZrO_3$ | 290 | 46.9 | 14 | 45.3 | 598 | 0.0782 | 76 | 36 |
| $Hf_{0.5}Zr_{0.5}O_2$ | 448 | 10.9 | 4.9 | 10.8 | 3260 | 0.003 | 3 | 47 |
| $SrBi_2Ta_2O_9$ | 565 | 2.4 | 1.4 | 4.9 | 600 | 0.004 | 8 | 35 |
| $CuInP_2S_6$ | 315 | 5.8 | 1.8 | 3.3 | 142 | 0.041 | 23 | 37 |
| 0.87PMN-0.13PT | 343 | 0.6 | 0.2 | 0.6 | 24 | 0.025 | 25 | 48 |
| BNBT/BCZT bilayer | 370 | 26.1 | 0.6 | 23 | 620 | 0.042 | 37 | 49 |
| P(VDF-TrFE) | 353 | 60 | 21 | 12.5 | 2090 | 0.0287 | 6 | 10 |
| $NaKC_4H_4O_6\cdot4H_2O$ | 265–298 | – | – | 0.003 | 1.2 | – | 25 | 50 |
| TGS | 323 | – | – | 0.11 | 1.6 | – | 68 | 16 |
| $NH_4HSO_4$ | 271 | 0.1 | 0.03 | 0.025 | 1.5 | 0.067 | 17 | 51 |
| $(iBA)_2PbCl_4$ | 302 | 25.64 | 7.7 | 11.06 | 29.7 | 0.86 | 370 | Herein |

Isothermal entropy change $|\Delta S|$, adiabatic temperature change $|\Delta T|$, and isothermal heat $|Q|$, at starting temperature $T$, due to changes of electric field $|\Delta E|$. For all entries, $Q = T\Delta S$.
PMN PbMg$_{1/3}$Nb$_{2/3}$O$_3$, PT PbTiO$_3$, P(VDF–TrFE) poly(vinylidene fluoride-trifluoroethylene) 55/45 mol%, TGS (NH$_2$CH$_2$COOH)3H$_2$SO$_4$, BNBT (Bi$_{0.5}$Na$_{0.5}$)TiO$_3$–BaTiO$_3$, BCZT Ba(Zr$_{0.2}$Ti$_{0.8}$)O$_3$–(Ba$_{0.7}$Ca$_{0.3}$)TiO$_3$.

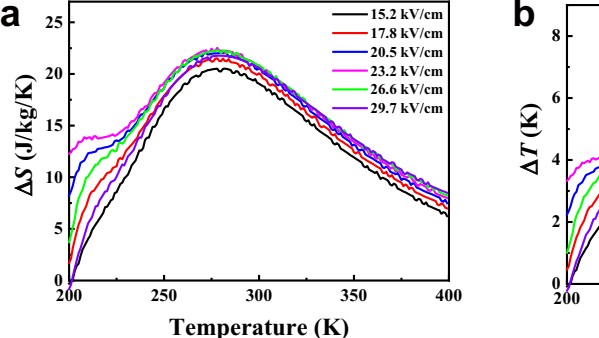
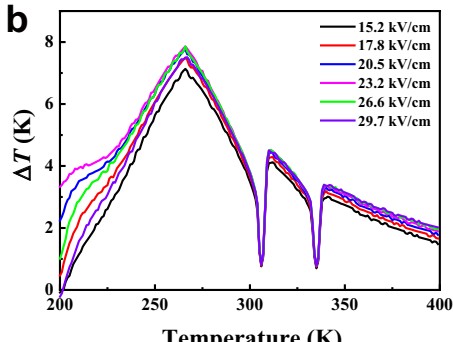

**Fig. 5 Ab initio calculated EC performance.** The EC entropy change $\Delta S$ (**a**) and the EC temperature change $\Delta T$ (**b**) for **1** under different electric field changes.

surfaces (Supplementary Fig. 13). The dielectric analyses were performed on the TongHui TH2828 analyzer with a scan rate of 10 K/min in the range of 275–355 K at the frequency of 100, 300, 500 kHz, and 1 MHz, respectively. The pyroelectric currents were measured using a high precision electrometer (Keithley 6517B). The temperature-dependent $P$–$E$ hysteresis loops were obtained by a ferroelectric analyzer (TF Analyzer 2000 FE-Module, aixACCT, Aachen, Germany) using a 1.5 mm-thick crystal sample. The direction of poling electric field is along **a** axis of the ferroelectric phase, which is determinate by a Supernova X-ray diffractometer combined with a polarizing microscope using a well-grown single crystal. In order to avoid electric discharge at high electric field, single crystals were immersed in silicone oil to measure the $P$–$E$ hysteresis loops.

The EC effect is evaluated by indirect method on the basis of Maxwell's relations[40]. The EC adiabatic temperature, $\Delta T$, and isothermal entropy, $\Delta S$ changes can be estimated indirectly as

$$\Delta T = -\frac{T}{C_p\rho}\int_0^{E_{max}}\left(\frac{\partial P}{\partial T}\right)_E \mathrm{d}E \qquad (2)$$

$$\Delta S = -\frac{1}{\rho}\int_0^{E_{max}}\left(\frac{\partial P}{\partial T}\right)_E \mathrm{d}E \qquad (3)$$

where $C_p$ is the heat capacity (1.22 J/g/K at 290 K); $\rho$ is density (1.925 g/cm$^3$); $E_{max}$ is maximum applied electric field; $P$ is ferroelectric polarization; The values of $(\partial P/\partial T)_E$ were deduced by seventh-order polynomial fitting of the temperature-dependent polarization curves. Computational investigation by using a CI-NEB-calculations-based thermodynamics description (details in Supporting Information), was employed for quantitative prediction of EC performances.

## Data availability

All relevant data are presented via this publication and Supplementary Information. The X-ray crystallographic coordinates for structures reported in this study have been deposited at the Cambridge Crystallographic Data Centre (CCDC), under deposition

numbers: CCDC 2080205, 2080206 and 2080207. These data can be obtained free of charge from The Cambridge Crystallographic Data Centre via www.ccdc.cam.ac.uk/getstructures. The data that support this study are available from the corresponding author upon reasonable request.

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

## Acknowledgements

This work was financially supported by the National Natural Science Foundation of China (21833010, 21875251, 21971238, 21975258, 61975207, 22075285, and 21921001), the Key Research Program of Frontier Sciences of the Chinese Academy of Sciences (ZDBS-LY-SLH024), the Strategic Priority Research Program of the Chinese Academy of Sciences (XDB20010200), the Youth Innovation Promotion of Chinese Academy of Sciences (2020307), the Natural Science Foundation of Fujian Province (2020J01112) and the Young Talent Supporting Project of Fujian Association of Science and Technology (2020000187).

## Author contributions

X. Liu, Z.W. and X. Li prepared the samples and carried out the experiments; P.L. did the FE measurements; T.G., H.J., and S.C. carried out the ab initio calculations; C.J. contributed to results analysis and manuscript writing; X. Liu, Z.S., and J.L. conceived the idea and co-supervised the research; X. Liu wrote the paper and all authors discussed the results and commented on the manuscript.

## Competing interests

The authors declare no competing interests.
