## [Peer Review File · Nature Communications]

REVIEWER COMMENTS

Reviewer #1 (Remarks to the Author):

In this manuscript, the authors reported a large electrocaloric effect near room temperature in the emerging family of hybrid perovskite ferroelectric. The phase transition behavior is measured by DSC, and dielectric vs. temperature. Ferroelectric hysteresis and electrocaloric performance were presented. These results are very interesting and important, which provide a family of candidate materials with potentially prominent electrocaloric performance for room temperature solid-state refrigeration. Therefore, it is recommended for publication in Nature Communication after minor revision.

(1) In Figure 1a, the authors should label the direction of ferroelectric polarization and the positive and negative ion centers.

(2) Bulk high quality crystals are very important for ferroelectric and electrocaloric measurements. The crystal growth process is too brief, more details are needed for others to reproduce the synthesis and crystal growth.

(3) The entropy change (ΔS) obtained by the Maxwell relation (25.64 J/(kg·K)) is smaller than that of integration of heat flow (30.26 J/(kg·K)), what's the reason?

Reviewer #2 (Remarks to the Author):

In this work the authors report on the observation of a giant electrocaloric response at room temperature in a novel ferroelectric material $[(\text{CH}_3)_2\text{CHCH}_2\text{NH}_3]_2\text{PbCl}_4$ having an organic-inorganic hybrid layered perovskite structure. While similar or higher electrocaloric temperature changes (ΔT) near room temperature have been reported in the past in several traditional ferroelectric perovskites (e.g. PbZrO_3 ($\Delta T = 30\text{K}$) (J. Mater. Chem. C 2018, 6, 10332); PMN-PT ($\Delta T = 9\text{K}$) or PbBaZrO_3 ($\Delta T = 45\text{K}$) (Peng et al., Adv. Funct. Mater. 2013, 23, 2987) mostly in thin or thick film structures, they have been calculated using the indirect methods. The reports of such high ΔT as predicted using the indirect methods have resulted in a growing scepticism in the field of such high temperature change values, since none of these high values have been experimentally validated via direct measurements. The indirect method is proven to have several inaccuracies and often leads to overestimation of ΔT if not applied with discretion. Nevertheless, the work presented here is interesting in terms of materials research and could have implications towards solid-state cooling applications using electrocaloric materials. The central claim is that the layered perovskite $[(\text{CH}_3)_2\text{CHCH}_2\text{NH}_3]_2\text{PbCl}_4$ exhibits a first-order ferroic phase transition at 302 K with a large associated latent heat (9140 J/kg) which imparts giant electrocaloric adiabatic temperature change of $\Delta T = 11\text{K}$ under nominal electric field changes of $\Delta E = 30\text{ kV/cm}$ as calculated via the indirect methods.

After review the paper, I find that there are several inadequacies in the paper which prevents its publication in Nature Communications in its current form. I recommend the following major corrections as listed below.

1. The authors claim that at the material $[(\text{CH}_3)_2\text{CHCH}_2\text{NH}_3]_2\text{PbCl}_4$ undergoes a ferroelectric-paraelectric (FE-PE) phase transition at $T_1 = 302$ K as revealed by the peaks in the heat flow (i.e. DSC measurements) and the step-like feature in the dielectric permittivity as in Fig. 2. However, it is not clear why T_1 is considered as the FE-PE Curie temperature. If it is indeed the Curie temperature, then the dielectric permittivity should obey the Curie-Weiss law in the PE phase after $T_1=302$. The authors should show the Curie-Weiss law fitting of the dielectric permittivity data in Fig. 2. It is not evident from the data that T_1 is a first-order phase transition in the material. Further, if $T_1 = 302$ K is the Curie temperature, what is the nature of the phase transition occurring at $T_2 = 335$ K? This needs to be explained further.

2. The dielectric loss is not provided for the material in Fig. 2 (b) which is crucial to estimate the features shown in the dielectric permittivity data. Please provide this data.

3. The authors claim that they have large cooling efficiency in the material, however, they also show a large thermal hysteresis accompanying the first-order phase transition which will affect the overall cooling efficiency of the device. This will greatly reduce the usability of these materials for refrigeration. Can you comment on this issue?

4. Considering the current status in the field of caloric cooling, there is a growing consensus on the use of direct measurements of adiabatic temperature change or isothermal entropy change to show caloric effects. This is mainly due to the inadequacies of using the indirect method using Maxwell's relations in case of certain ferroic phase transitions. It is recommended that the authors show these giant electrocaloric effects via direct measurement of adiabatic temperature change and not rely on indirect methods to justify publication in Nature Communications. The efficacy of the indirect measurements needs to be justified to warrant publication. There are several assumptions such as field-independent heat capacity used in the indirect methods which can give erroneous results and overestimation of caloric effects.

5. In order to calculate the adiabatic temperature change using the indirect method as in Fig. 4, the value of the heat capacity is required. The authors do not provide this data in the paper. How did you calculate the heat capacity?

6. It is quite clear from Fig. 2 that the values of the dielectric constant across T_1 are quite low (≈ 4) as compared to inorganic perovskites like BaTiO_3 or PZT ($\approx 3000-10,000$). Further, the values of the dielectric constants increase with increasing frequency which is contrary to traditional dielectric materials. This difference in dielectric behaviour needs to be justified. Due to the low value of the dielectric constant, the material $[(\text{CH}_3)_2\text{CHCH}_2\text{NH}_3]_2\text{PbCl}_4$ should have low threshold for dielectric breakdown as compared to traditional dielectric materials. The authors need to provide the

measurements of the electric fields required for dielectric breakdown in this material in order to claim that this material gives such high electrocaloric strengths at such low applied electric field changes ($\Delta E = 30 \text{ kV/cm}$). This is an important point since the high electrocaloric effects are observed primarily in thin film structures since one can drive higher electrocaloric effects in thin films by applying higher electric field changes (i.e. see Table 1 refs. 9, 35, 40, 34, 42, 10) due to their higher threshold for dielectric breakdown. However, here, the authors get higher electrocaloric effects at much lower fields. This point has to be justified.

7. The remnant polarization in the material ($5.2 \mu\text{C/cm}^2$) is lower than most traditional ferroelectric perovskites. It is not clear to me why with such low polarization values the authors observe such high electrocaloric effects which are much higher than those obtained for inorganic ferroelectric perovskite such as BaTiO_3 , PMN-PT, PZT obtained under the similar applied fields (see Table 1) . Please justify. What maximum voltages were used during the polarization measurements shown in Fig. 4(a)? What was the thickness of the crystal that was poled and how was that estimated?

8. The paper lacks any microstructural analysis on the single crystals. Some microscopic images of the crystals and the fabricated ferroelectric capacitor using the single crystal should be provided.

9. Some irrelevant data has been provided in the supporting information which have no reference in the manuscript such as Figs. S4, S5, S6, S7, S8. Justify these figures in the Supporting Information.

10. The nature of the temperature dependent polarization $P(T)$ curves shown in Fig. 3 (a) as obtained from the pyroelectric measurements and in Fig. 4 (b) as obtained from the polarization measurements are drastically different. While the $P(T)$ curve in Fig. 3(a) shows a sharp transition at T_1 , the $P(T)$ curves in Fig. 4 (b) do not show any such sharp transition. Please account for the difference in the natures of these graphs.

11. Leakage is a major source of errors in the polarization measurements using the integration of pyroelectric current. How was leakage effects in the ferroelectric capacitors taken into effect during these measurements?

12. The authors ascribe the quantity $\Delta T/\Delta E$ as the cooling performance as shown in Fig. 4 (e). However, it is not justified as to why the quantity is being described as the cooling performance. Typically, the cooling efficiency of an electrocaloric material is determined by the refrigeration capacity (RC) or the materials efficiency [Ref. 3 in the manuscript]. Please justify this point and modify the Fig. 4 (e) by showing comparison of RC or energy efficacy of various EC materials.

My overall impression is that the work presented here is suited for a materials journal since it reports a higher electrocaloric response in a novel materials system. Neither the work provides any new underlying physics on electrocaloric materials or gives a novel idea for electrocaloric device fabrication which can advance the field and can be considered justified to the scope of Nature Communications.

Reviewer #3 (Remarks to the Author):

Electrocaloric effect (ECE) based on ferroelectric materials has been intensively investigated in recent years as a candidate for the solid-state refrigeration devices. The organic-inorganic hybrid ferroelectrics exhibits exceptional structural diversity, large heat exchange and broad property tenability, which is a promising direction but has not yet been explored for ECE applications. Here, the authors investigated the ECE of layered hybrid perovskite ferroelectric (iBA)₂PbCl₄. By using the indirect method based on Maxwell relation, excellent performance of adiabatic temperature change and ECE strength $\Delta T/\Delta E$ has been realized around room temperature. The ECE strength $\Delta T/\Delta E$ of this (iBA)₂PbCl₄ single crystal is almost one order of magnitude larger than that of traditional ferroelectric materials, which means a large ECE can be achieved under a relatively small electric field. Although this (iBA)₂PbCl₄ single crystal also faces the limitation of narrow temperature span due to the sharp first-order transition, these results bring new prospects in the field of room temperature ECE refrigeration based on organic-inorganic hybrid perovskites. However, the authors should address the following points:

1. This work is an extension of the authors' previous work. [Adv. Funct. Mater. 2019, 29, 1805038], where the new organic-inorganic hybrid ferroelectric of (C₄H₉NH₃)₂PbCl₄ is firstly reported, and it exhibits intrinsic broadband white-light emission with high color rendering index. According to the DSC data informing the phase transition temperature and hysteresis, XRD peak intensities, as well as the temperature dependent dielectric curves, it seems that the single crystal of (iBA)₂PbCl₄ in this present work is different from the previous one [AFM2019], could the authors explain the reason? I notice that there are small differences of the processing parameter like cooling rate and excess ratio of isobutylamine cation, are these determined factors? If not, does it mean that the fabrication of (iBA)₂PbCl₄ is not controllable to obtain the same phase transition? Anyway, the authors should cite this AFM paper, because this (C₄H₉NH₃)₂PbCl₄ was firstly reported there.

2. Page 13, line 235, the indirect method based on Maxwell relation is employed to demonstrate the ECE of (iBA)₂PbCl₄ in this work, but the detailed value of heat capacity is not given in the manuscript. Also, if the initial electric field is selected to be zero, the authors are encouraged to give P-T curves at lower electric field for Fig. 4b, or maybe put it in the supporting information as the substitute. The frequency of the applied electric field determines the shape of ferroelectric hysteresis loop which is also important for further operation of real ECE cooling cycles but not given in this manuscript. According to Fig. 3a and Fig. S6, the hysteresis loops show that the maximum polarization is not obtained under the largest electric field, which may result from leakage, ion migration and defects. The authors are encouraged to give the dielectric loss under the dielectric curves in Fig. 2b, also the direct method are suggested to be employed to test the ECE response of the system, which will make the results more convincing.

3. It's well known that the organic-inorganic hybrid perovskite is sensitive to the atmosphere like humidity and other contamination in the air, which are adverse factors of this material for real application of long term and stable cooling cycles. The authors are encouraged to give the stability of

this $(iBA)_2PbCl_4$ both environmentally and electrically.

1)The authors have given more data in the Supplementary Information, but it has not been detailed discussed from Fig. S3 to Fig. S8. Sometimes, readers could not understand what is the meaning without explanation .

Some minor comments.

1)Page 4, line 75, Fig. 1 what is the meaning of arrows, it should be demonstrated in the title or in the main text.

2)Page 7, line 118, T_c is written in a different font from others. The authors are encouraged to double check the main text.

3)Page 7, line 131, polarization is represented as P_s . At 293 K, Page 9, line 151, At 293 K, the P_s value is derived to be $5.2 \mu C/cm^2$ with a coercive electric field of 15 kV/cm.

It really makes me confused that this P_s is the remnant polarization, spontaneous polarization or maximum polarization? If it is the remnant polarization, that value should be lower than 5 according to the figure 3a. Again, Page 13, line 235, P is called polarization.

4)Page 9, line 151, "...(Supporting information)." should be "Methods"?

5)Page 13, line 225, the thickness of your samples used for testing P-E hysteresis loops should be especially demonstrated here, which has influence of the P-E hysteresis loops due to the concentration of defects or other thickness related issues.

Response to Reviewers

Reviewer: 1

Comments:

In this manuscript, the authors reported a large electrocaloric effect near room temperature in the emerging family of hybrid perovskite ferroelectric. The phase transition behavior is measured by DSC, and dielectric vs. temperature. Ferroelectric hysteresis and electrocaloric performance were presented. These results are very interesting and important, which provide a family of candidate materials with potentially prominent electrocaloric performance for room temperature solid-state refrigeration. Therefore, it is recommended for publication in Nature Communication after minor revision.

➤ **Response:** Thanks for the reviewer's valuable comments on our manuscript. According to the reviewer's suggestion, we have revised the manuscript accordingly.

(1) In Figure 1a, the authors should label the direction of ferroelectric polarization and the positive and negative ion centers.

➤ **Response:** Thank you for reviewer's suggestion. The direction of ferroelectric polarization and the positive and negative ion centers have been added in **Figure 1** of the revised manuscript.

Figure 1 | Illustration of phase transitions of 1. **a**, Monoclinic phase at 285 K. **b**, Orthorhombic phase at 318 K. **c**, Orthorhombic phase at 343 K. The blue arrow in **a** presents the direction of spontaneous polarization. The red and pink arrows in **b** and **c** demonstrate the rotation of iso-butylammonium (iBA) cations.

(2) Bulk high quality crystals are very important for ferroelectric and electrocaloric

measurements. The crystal growth process is too brief, more details are needed for others to reproduce the synthesis and crystal growth.

- **Response:** According to the reviewer's helpful suggestion, we have described the experimental synthesis in detail. The related contents have been added in the revision, which is benefit for others to reproduce the synthesis and crystal growth. Please see the Experimental Section:

*"Bulk single crystals were grown from a saturated solution by the temperature lowering method with a cooling rate of 0.5 °C/day. The saturated solution of **1** were prepared at about 50°C and then kept 4~6°C above its preliminary saturated temperature for 10 hours, which can ensure the dissolution of all ingredients. Then, the saturated solution were slowly cooling with a temperature rate of 0.2 K/day. Finally, the pristine crystals with dimensions up to 15 × 5 × 0.5 mm³ (**Figure S1**), which were wiped and dried in a N₂ glovebox overnight, and annealed at 70 °C for 2 h to completely remove moisture and release lattice stress."*

Figure S1. The grown single crystals of $[(\text{CH}_3)_2\text{CHCH}_2\text{NH}_3]_2\text{PbCl}_4$

- (3) The entropy change (ΔS) obtained by the Maxwell relation (25.64 J/(kg·K)) is smaller than that of integration of heat flow (30.26 J/(kg·K)), what's the reason?
- **Response:** Thanks for the reviewer's valuable comment. These values are smaller than the zero-field DSC measured value of 30.26 J/(kg·K) at T_C , implying possible contributions to the total entropy change from other instabilities such as the rotation of PbCl_4 octahedra.

Reviewer: 2

Comments:

In this work the authors report on the observation of a giant electrocaloric response at room temperature in a novel ferroelectric material $[(\text{CH}_3)_2\text{CHCH}_2\text{NH}_3]_2\text{PbCl}_4$ having an organic-inorganic hybrid layered perovskite structure. While similar or higher electrocaloric temperature changes (ΔT) near room temperature have been reported in the past in several traditional ferroelectric perovskites (e.g. PbZrO_3 ($\Delta T = 30\text{K}$) (J. Mater. Chem. C 2018, 6, 10332); PMN-PT ($\Delta T = 9\text{K}$) or PbBaZrO_3 ($\Delta T = 45\text{K}$) (Peng et al., Adv. Funct. Mater. 2013, 23, 2987) mostly in thin or thick film structures, they have been calculated using the indirect methods. The reports of such high ΔT as predicted using the indirect methods have resulted in a growing scepticism in the field of such high temperature change values, since none of these high values have been experimentally validated via direct measurements. The indirect

method is proven to have several inaccuracies and often leads to overestimation of ΔT if not applied with discretion.

Nevertheless, the work presented here is interesting in terms of materials research and could have implications towards solid-state cooling applications using electrocaloric materials. The central claim is that the layered perovskite $[(\text{CH}_3)_2\text{CHCH}_2\text{NH}_3]_2\text{PbCl}_4$ exhibits a first-order ferroic phase transition at 302 K with a large associated latent heat (9140 J/kg) which imparts giant electrocaloric adiabatic temperature change of $\Delta T = 11$ K under nominal electric field changes of $\Delta E = 30$ kV/cm as calculated via the indirect methods.

➤ **Response:** We deeply thank the reviewer for the time in reviewing the paper and professional feedback. The questions and suggestions are very critical to improve our work. We have tried our best to address the questions and revised our manuscript accordingly.

(1) The authors claim that at the material $[(\text{CH}_3)_2\text{CHCH}_2\text{NH}_3]_2\text{PbCl}_4$ undergoes a ferroelectric-paraelectric (FE-PE) phase transition at $T_1 = 302$ K as revealed by the peaks in the heat flow (i.e. DSC measurements) and the step-like feature in the dielectric permittivity as in Fig. 2. However, it is not clear why T_1 is considered as the FE-PE Curie temperature. If it is indeed the Curie temperature, then the dielectric permittivity should obey the Curie-Weiss law in the PE phase after $T_1=302$. The authors should show the Curie-Weiss law fitting of the dielectric permittivity data in Fig. 2. It is not evident from the data that T_1 is a first-order phase transition in the material. Further, if $T_1 = 302$ K is the Curie temperature, what is the nature of the phase transition occurring at $T_2 = 335$ K? This needs to be explained further.

➤ **Response:** Thanks for the reviewer's professional comments. This is an interesting but complex work to study the ferroelectric-paraelectric (FE-PE) phase transition of **1**. From the results of DSC and temperature dependence dielectric permittivity, the phase transition at $T_2 = 335$ K mostly like the T_C , due to the good agreement of the Curie-Weiss law of the dielectric anomalies (Fig. S10). While, the temperature dependent SHG response (Fig. 3b) reveal a non-centrosymmetric (polar FE phase) to centrosymmetric (nonpolar PE phase) structure transition occur at T_i , suggesting that T_i is mostly likely the T_C . Furthermore, the most striking evidence of the T_C is the results of pyroelectric (Fig. 3a) and P - E measurements (Fig. 4a and b), which show that the pyroelectric current and spontaneous polarization disappear over T_i , indicting the ferroelectric feature of the material disappears at T_i . Taking the above mentioned results, T_i is considered as the FE-PE Curie temperature of the materials. The step-like feature at T_i in the dielectric permittivity is proposed to originate from PbCl_6 octahedral rotation induced improper ferroelectricity, resembling that of layered perovskite oxide $(\text{Ca,Sr})_3\text{Ti}_2\text{O}_7$ (*Nat. Mater.* 2015, 14 (4), 407) and $\text{Ca}_3\text{Mn}_2\text{O}_7$ (*Phys. Rev. Lett.* 2011, 106 (10), 107204). Otherwise, we consider that the intermediate phase between T_i and T_2 may be an antiferroelectric phase. Much efforts have been done in our laboratory to measure the P - E loops under high electric fields. Unfortunately, it is failure to obtain the double hysteresis loops until the breakdown of the crystal samples (> 60 kV/cm). As you suggested, the Curie-Weiss law fitting of the dielectric permittivity and the possibility of the phase

transition at T_2 have been added in the revised manuscript.

Figure S10. The fitting to Curie-Weiss law of dielectric anomalies at 500 kHz.

- (2) The dielectric loss is not provided for the material in Fig. 2 (b) which is crucial to estimate the features shown in the dielectric permittivity data. Please provide this data.
- **Response:** As the reviewer referred, the dielectric loss is crucial to estimate the features of dielectric permittivity data. Taking the reviewer's suggestion, the missed temperature and frequency dependent dielectric loss data has been added in revised manuscript (Fig. 2b and S9).

Figure 2b. Temperature-dependent dielectric constant (ϵ) and dielectric loss ($\tan \delta$) of **1**.

Figure S9. Frequency-dependent dielectric constant (ϵ) and dielectric loss ($\tan \delta$) of **1**.

- (3) The authors claim that they have large cooling efficiency in the material, however, they also show a large thermal hysteresis accompanying the first-order phase transition which will affect the overall cooling efficiency of the device. This will greatly reduce the usability of these materials for refrigeration. Can you comment on this issue?
- **Response:** Thanks for the reviewer's professional comments. As you referred, large thermal hysteresis related to the first-order phase transition at T_c may raise concern on the reversibility of the EC effect, which can be eliminated in normal ferroelectric with the second-order phase transition and relaxors with the significantly diffused phase transition. Since the entropy changes associated to the second-order phase transition are relatively small, it is not appropriate for solid-state EC refrigeration. For the emerging hybrid ferroelectrics, the exceptional structural flexibility and chemical diversity make it possible to design relaxor solid solution through cations and halogen substitutions [*Science* 2019, 363 (6432), 1206; *Adv. Mater.* 2016, 28 (13), 2579], which will be an effective approach to eliminate the thermal hysteresis. In future work, we will devote more efforts to design relaxor hybrid ferroelectrics to solve the issue of thermal hysteresis accompanying the phase transitions.
- (4) Considering the current status in the field of caloric cooling, there is a growing consensus on the use of direct measurements of adiabatic temperature change or isothermal entropy change to show caloric effects. This is mainly due to the inadequacies of using the indirect method using Maxwell's relations in case of certain ferroic phase transitions. It is recommended that the authors show these giant electrocaloric effects via direct measurement of adiabatic temperature change and not rely on indirect methods to justify publication in Nature Communications. The efficacy of the indirect measurements needs to be justified to warrant publication. There are several assumptions such as field-independent heat capacity used in the indirect methods which can give erroneous results and overestimation of caloric effects.
- **Response:** Thanks a lot for the reviewer's precious comments and suggestions. As you referred, in present study, we mainly focused on original exploration and development

of EC materials through chemical design and aimed at develop new sources of ferroelectric materials which generating giant EC effect. In comparison to the most studied inorganic perovskite and organic polymer ferroelectrics, the research of hybrid perovskite ferroelectrics for EC refrigeration remains unprecedented, which could provide an ideal platform to design more advanced EC materials benefiting from their superior structural diversity, large heat exchange and broad property tenability.

It is known that indirect measurement, based on the Maxwell relations, has become a well-established practice in the EC research, which could provide general trends to analyze the behavior of EC responses and is useful for rapid selection of EC materials. The indirect measurement present in this work may not be optimal, but should be sufficient to reveal the EC ability of hybrid perovskite ferroelectrics. The direct measurements may be better, which are under investigation in our laboratory. Unfortunately, results are unavailable at this point. In other way, theoretical approach represents an important advance to quantify the EC effect by using ab initio calculations. To make up for the pity, we present a theoretical investigation based on first-principle effective Hamiltonian approach (Details in supporting information), which represents a very attractive and powerful method for the quantitative prediction of EC properties. As shown in Figure 5, computational isothermal entropy changes and adiabatic temperature changes based on the Landua-Ginzburg theory indicate that the EC performance of $(iBA)_2PbCl_4$ is almost one order magnitude larger than those of the traditional ferroelectrics, with an entropy change ΔS of about 26.9 J/kg/K and an adiabatic temperature change ΔT of about 7.3 K at room temperature under the applied electric field of 29.7 KV/cm, which is in consistent with the results evaluated by indirect method. Both indirect measurement and theoretical investigation indicate that the family of hybrid perovskite ferroelectric materials as the promising candidates for room temperature EC refrigeration.

In future work, we will devote more efforts to the construction of high-quality thin-film and large size single-crystal as well as seek cooperation to carry out the direct measurements and explore further practical application of hybrid perovskite ferroelectrics in cooling devices.

Figure 4 | Ab initio calculated EC performance. The EC entropy change ΔS **a)** and the EC temperature change ΔT **b)** for **1** under different electric field changes.

- (5) In order to calculate the adiabatic temperature change using the indirect method as in Fig. 4, the value of the heat capacity is required. The authors do not provide this data in the paper. How did you calculate the heat capacity?
- **Response:** Thanks for the reviewer's professional comments, the missing heat capacity has been added in the revised supporting information, which was conducted on a NETZSCH DSC 200 F3 DSC instrument with a heating and cooling rate of 10 K/min under nitrogen atmosphere. The sapphire sample was used as the standard to calculate the specific heat values.

Figure S4. Temperature-dependent specific heat (C_p) of **1**.

- (6) It is quite clear from Fig. 2 that the values of the dielectric constant across T_1 are quite low (≈ 4) as compared to inorganic perovskites like BaTiO_3 or PZT ($\approx 3000-10,000$). Further, the values of the dielectric constants increase with increasing frequency which is contrary to traditional dielectric materials. This difference in dielectric behaviour needs to be justified. Due to the low value of the dielectric constant, the material $[(\text{CH}_3)_2\text{CHCH}_2\text{NH}_3]_2\text{PbCl}_4$ should have low threshold for dielectric breakdown as compared to traditional dielectric materials. The authors need to provide the measurements of the electric fields required for dielectric breakdown in this material in order to claim that this material gives such high electrocaloric strengths at such low applied electric field changes ($\Delta E = 30 \text{ kV/cm}$). This is an important point since the high electrocaloric effects are observed primarily in thin film structures since one can drive higher electrocaloric effects in thin films by applying higher electric field changes (i.e. see Table 1 refs. 9, 35, 40, 34, 42, 10) due to their higher threshold for dielectric breakdown. However, here, the authors get higher electrocaloric effects at much lower fields. This point has to be justified.
- **Response:** Thanks a lot for the reviewer's precious comments. The dielectric properties including temperature and frequency dependent dielectric properties were remeasured, which shows that values of the dielectric constants are indeed quite low and decrease with increasing frequency. Inaccurate short-circuit calibration introduce

some error in our original data.

It is known that the relationship between dielectric breakdown strength (E_{bd}) and dielectric constant (k) is quite complex. The most convincing model reported in literature [*Appl. Phys. Lett.* 2003, 82 (13), 2121; *International Electron Devices Meeting*, IEEE: 2002; 633] shows that breakdown strength exhibits an approximate $(k)^{-1/2}$ dependence over a wide range of dielectric materials. To date, a lot of work has proved that breakdown strength increases with a decrease in dielectric constant for the same material type. Table R1 and Figure R1 present some results of dielectric breakdown strength and dielectric constant of some known dielectrics [*Appl. Phys. Lett.* 2003, 82 (13), 2121]. The dielectric constant of our sample is indeed quite low (≈ 4) as compared to inorganic perovskites, suggesting that the material should have high (not low) threshold for dielectric breakdown. However, we carried out the breakdown strength measurement and the results show that the material breaks down at the surface at about 60 KV/cm at room temperature. This may be caused by the high degree of defects at the crystal surface during the crystal growth. In order to get higher EC effect and promote their applications, bulk high quality single crystals or films are very important.

[Redacted]

[Redacted]

Figure R1. Relationship between dielectric breakdown strength and dielectric constant in some high dielectric constant materials reported in literature [*Appl. Phys. Lett.* 2003, 82 (13), 2121].

- (7) The remnant polarization in the material ($5.2 \mu\text{C}/\text{cm}^2$) is lower than most traditional ferroelectric perovskites. It is not clear to me why with such low polarization values the authors observe such high electrocaloric effects which are much higher than those obtained for inorganic ferroelectric perovskite such as BaTiO_3 , PMN-PT, PZT obtained under the similar applied fields (see Table 1) . Please justify. What maximum voltages were used during the polarization measurements shown in Fig. 4(a)? What was the thickness of the crystal that was poled and how was that estimated?
- **Response:** Thanks for the reviewer's professional comments. The EC effect in **1** is evaluated by indirect method based on the Maxwell relations,
$$\Delta T = -\frac{T}{c_p \rho} \int_0^{E_{max}} \left(\frac{\partial P}{\partial T} \right)_E dE.$$
 Despite the low polarization in the material, it is worth noticing that a fast polarization change over 70% at T_1 occurs within only 3 K (**Fig. 4b**), resulting in a significant $(\partial P/\partial T)_E$ value, which is associated to the sharp first-order phase transition. In addition, the driving electric field required to generate a sharp phase transition is also lower than that of most traditional ferroelectric perovskites. The underlying mechanism of the prominent EC performance is attributed to the large polarization change and relatively small driving electric field required to generate a sharp phase transition. Although, the magnitude of EC effect around T_C is significantly larger than that of most traditional ferroelectric perovskites (e.g. BaTiO_3 , PMN-PT, PZT), the working temperature window becomes narrower using the first-order phase transition. In addition, the maximum voltage used during the polarization measurements is 4500 V and the thickness of the crystal is about 1.5 mm, which was measured under an optical microscope.
- (8) The paper lacks any microstructural analysis on the single crystals. Some microscopic images of the crystals and the fabricated ferroelectric capacitor using the single crystal should be provided.

- **Response:** Thanks for the reviewer's valuable suggestions. The microstructural analysis on the single crystals have been added in the revised manuscript. The obtained single crystals were transparent colorless plate like sheets with dimensions of millimeter scale. Observation under a polarizing microscope confirmed that the grown single crystals are in the single domain state. Pieces of crystal wafers were cut perpendicular to the [100] crystallographic direction from a well-grown single crystal and coated with silver electrode on both sides were used for electrical measurements. The direction was determinate by a Supernova X-ray diffractometer combined with a polarizing microscope.

Figure S1. The grown single crystals of $[(\text{CH}_3)_2\text{CHCH}_2\text{NH}_3]_2\text{PbCl}_4$

Figure S3. Crystal morphology of grown crystals. Inset: fabricated ferroelectric capacitor.

- (9) Some irrelevant data has been provided in the supporting information which have no reference in the manuscript such as Figs. S4, S5, S6, S7, S8. Justify these figures in the Supporting Information.
- **Response:** Thanks for the reviewer's valuable suggestions. All figures of Supporting Information have been justified in the revised manuscript.
- (10) The nature of the temperature dependent polarization $P(T)$ curves shown in Fig. 3 (a) as obtained from the pyroelectric measurements and in Fig. 4 (b) as obtained from the polarization measurements are drastically different. While the $P(T)$ curve in Fig. 3(a) shows a sharp transition at T_1 , the $P(T)$ curves in Fig. 4 (b) do not show any such sharp transition. Please account for the difference in the natures of these graphs.
- **Response:** Thanks for the reviewer's professional comments. The drastically difference between Fig. 3 (a) and Fig. 4 (b) is mainly caused by the displayed temperature range.

When enlarged the Fig. 3(a), it is worth noticing that a fast polarization change over 80% is observed within only 3 K, which is consistent with Fig. 4 (b). Meanwhile, the pyroelectric currents were measured at zero-bias, while the P-E loops were measured at high electric fields. Under high electric field, the increases of conductivity of the material as the temperature rises may also introduce some error in Fig. 4 (b).

(11) Leakage is a major source of errors in the polarization measurements using the integration of pyroelectric current. How was leakage effects in the ferroelectric capacitors taken into effect during these measurements?

➤ **Response:** Thank you for reviewer's professional suggestion. As you referred, leakage is a major source of errors in the polarization measurements. While the pyroelectric measurements were carried out at zero-bias, the leakage currents are very small (~pA) in compared to pyroelectric currents (~nA), as shown in inset of Fig. 3(a).

(12) The authors ascribe the quantity $\Delta T/\Delta E$ as the cooling performance as shown in Fig. 4 (e). However, it is not justified as to why the quantity is being described as the cooling performance. Typically, the cooling efficiency of an electrocaloric material is determined by the refrigeration capacity (RC) or the materials efficiency [Ref. 3 in the manuscript]. Please justify this point and modify the Fig. 4 (e) by showing comparison of RC or energy efficacy of various EC materials.

➤ **Response:** Thank you for reviewer's professional suggestion. The comparison of RC and energy efficacy of various EC materials have been added in revised manuscript (Fig. 4e).

Reviewer: 3

Comments:

Electrocaloric effect (ECE) based on ferroelectric materials has been intensively investigated in recent years as a candidate for the solid-state refrigeration devices. The organic-inorganic hybrid ferroelectrics exhibits exceptional structural diversity, large heat exchange and broad property tenability, which is a promising direction but has not yet been explored for ECE applications. Here, the authors investigated the ECE of layered hybrid perovskite ferroelectric $(iBA)_2PbCl_4$. By using the indirect method based on Maxwell relation, excellent performance of adiabatic temperature change and ECE strength $\Delta T/\Delta E$ has been realized around room temperature. The ECE strength $\Delta T/\Delta E$ of this $(iBA)_2PbCl_4$ single crystal is almost one order of magnitude larger than that of traditional ferroelectric materials, which means a large ECE can be achieved under a relatively small electric field. Although this $(iBA)_2PbCl_4$ single crystal also faces the limitation of narrow temperature span due to the sharp first-order transition, these results bring new prospects in the field of room temperature ECE refrigeration based on organic-inorganic hybrid perovskites.

➤ **Response:** Thank the reviewer for the kind comments. We are glad to revise our manuscript according to your valuable concerns.

(1) This work is an extension of the authors' previous work. [Adv. Funct. Mater. 2019, 29, 1805038], where the new organic-inorganic hybrid ferroelectric of $(C_4H_9NH_3)_2PbCl_4$ is

firstly reported, and it exhibits intrinsic broadband white-light emission with high color rendering index. According to the DSC data informing the phase transition temperature and hysteresis, XRD peak intensities, as well as the temperature dependent dielectric curves, it seems that the single crystal of (iBA)₂PbCl₄ in this present work is different from the previous one [AFM2019], could the authors explain the reason? I notice that there are small differences of the processing parameter like cooling rate and excess ratio of isobutylamine cation, are these determined factors? If not, does it mean that the fabrication of (iBA)₂PbCl₄ is not controllable to obtain the same phase transition? Anyway, the authors should cite this AFM paper, because this (C₄H₉NH₃)₂PbCl₄ was firstly reported there.

- **Response:** Thanks for the reviewer's valuable comments on our manuscript. As reviewer's referred, in our previous work published in *Adv. Funct. Mater.* **2019**, **29**, **1805038**, we reported a 2D perovskite-type ferroelectric (n-butylamine)₂PbBr₄. While, the compound reported [AFM2019] is not the same as the one reported in this work. The main difference between two compounds is the organic spacer cations residing between the inorganic perovskite frameworks (Figure R2). The spacer cation reported in [AFM2019] is n-butylammonium (nBA), while the spacer cation reported here is iso-butylammonium (iBA) (Figure R3). The following table (Table R1) also presents the differences of the crystallographic parameters of the two compounds. In addition, the corresponding reference has been added in the revised manuscript.

Figure R2. The spacer cation of n-butylammonium (nBA) and iso-butylammonium (iBA).

Figure R3. Crystal structures of $(\text{C}_4\text{H}_9\text{NH}_3)_2\text{PbCl}_4$ and $[(\text{CH}_3)_2\text{CHCH}_2\text{NH}_3]_2\text{PbCl}_4$.

Table R1. Crystal data for 1 collected at 200 K, 298 K and 400 K.

Empirical formula	$[\text{CH}_3(\text{CH}_2)_3\text{NH}_3]_2\text{PbCl}_4$	$[(\text{CH}_3)_2\text{CHCH}_2\text{NH}_3]_2\text{PbCl}_4$
Formula weight	497.28	497.28
Temperature (K)	216	285
Space group	$Cmc2_1$	Pm
Cell parameters	$a = 27.8521(6)\text{\AA}$	$a = 7.9019(3)\text{\AA}$
	$b = 7.968(2)\text{\AA}$	$b = 27.4789(15)\text{\AA}$
	$c = 7.788(2)\text{\AA}$	$c = 7.9019(3)\text{\AA}$
	$\alpha = 90^\circ$	$\alpha = 90^\circ$
	$\beta = 90^\circ$	$\beta = 90.0000(10)^\circ$
	$\gamma = 90^\circ$	$\gamma = 90^\circ$
$V (\text{\AA}^3)$	1728.4(7)	1715.78(13)
$Z, \rho_{\text{cal.}} (\text{g/cm}^3)$	4, 1.911	4, 1.925

(2) Page 13, line235, the indirect method based on Maxwell relation is employed to demonstrate the ECE of $(\text{iBA})_2\text{PbCl}_4$ in this work, but the detailed value of heat capacity is not given in the manuscript. Also, if the initial electric field is selected to be

zero, the authors are encouraged to give P-T curves at lower electric field for Fig. 4b, or maybe put it in the supporting information as the substitute. The frequency of the applied electric field determines the shape of ferroelectric hysteresis loop which is also important for further operation of real ECE cooling cycles but not given in this manuscript. According to Fig. 3a and Fig. S6, the hysteresis loops show that the maximum polarization is not obtained under the largest electric field, which may result from leakage, ion migration and defects. The authors are encouraged to give the dielectric loss under the dielectric curves in Fig. 2b, also the direct method are suggested to be employed to test the ECE response of the system, which will make the results more convincing.

- **Response:** Thanks for the reviewer's professional comments. As the reviewer referred, the missing heat capacity (Fig. S4), *P-T* curves at lower electric field (Fig. S12) and frequency of the applied electric field (50 Hz) have been added in the revised manuscript. The dielectric loss is crucial to estimate the features of dielectric permittivity data. Taking the reviewer's suggestion, the dielectric loss data also has been added in revised manuscript (Fig. 2b).

It is known that indirect measurement, based on the Maxwell relations, has become a well-established practice in the EC research, which could provide general trends to analyze the behavior of EC responses and is useful for rapid selection of EC materials. The indirect measurement present in this work may not be optimal, but should be sufficient to draw our conclusions. The direct measurements may better reveal the EC ability of these materials. The direct measurements are also under investigation in our laboratory. Unfortunately, results are unavailable at this point. In other way, the theoretical approach represents an important advance to quantify the EC effect by using ab initio calculations. To make up for the pity, we designed the present study to and we had some interesting findings. (More information can refer to the Q4 of reviewer #2 and supporting information)

Figure S4. Temperature-dependent specific heat (C_p) of 1.

- (3) It's well known that the organic-inorganic hybrid perovskite is sensitive to the atmosphere like humidity and other contamination in the air, which are adverse factors of this material for real application of long term and stable cooling cycles. The authors are encouraged to give the stability of this $(\text{iBA})_2\text{PbCl}_4$ both environmentally and electrically.
- **Response:** Thanks a lot for the constructive suggestion. The phase stability was characterized by the powder X-ray diffraction (PXRD). As shown in Fig. S3, **1** exhibits good phase stability after exposing to air for 100 days. Meanwhile, the electric polarization of **1** could keep stable without any obvious fatigue after approximately 2×10^7 switching cycles between ON and OFF states. These results show that the crystal shows good stability both environmentally and electrically.

Figure S3. Powder X-ray diffractions patterns of **1** in the different test conditions.

Figure S4. Variation of polarization versus number of switching cycles of **1**.

The authors have given more data in the Supplementary Information, but it has not been detailed discussed from Fig. S3 to Fig. S8. Sometimes, readers could not understand what is

the meaning without explanation.

- **Response:** Thanks for the reviewer's valuable suggestions. All figures of Supporting Information have been justified in the revised manuscript.

Some minor comments.

- (1) Page 4, line 75, Fig. 1 what is the meaning of arrows, it should be demonstrated in the title or in the main text.

- **Response:** Thanks for the reviewer's valuable comments, the meaning of arrows have been demonstrated in the title (**Figure 1**).

Figure 1 | Illustration of phase transitions of 1. a, Monoclinic phase at 285 K. b, Orthorhombic phase at 318 K. c, Orthorhombic phase at 343 K. The blue arrow in **a** presents the direction of spontaneous polarization. The red and pink arrows in **b** and **c** demonstrate the rotation of iso-butylammonium (iBA) cations.

- (2) Page 7, line 118, T_c is written in a different font from others. The authors are encouraged to double check the main text.

- **Response:** Thanks for the reviewer's valuable comments, the wrong written have been well check and corrected in the revised manuscript.

- (3) Page 7, line 131, polarization is represented as P_s. At 293 K, Page 9, line 151, At 293 K, the P_s value is derived to be 5.2 μC/cm² with a coercive electric field of 15 kV/cm.

It really makes me confused that this P_s is the remnant polarization, spontaneous polarization or maximum polarization? If it is the remnant polarization, that value should be lower than 5 according to the figure 3a. Again, Page 13, line 235, P is called polarization.

- **Response:** Thanks for the reviewer's valuable comments, the polarization used in the text were checked and revised in the revised manuscript.

- (4) Page 9, line 151, "...(Supporting information)." should be "Methods" ?

- **Response:** Thanks for the reviewer's valuable comments, the wrong written have

been corrected in the revised manuscript.

- (5) Page 13, line 225, the thickness of your samples used for testing P-E hysteresis loops should be especially demonstrated here, which has influence of the P-E hysteresis loops due to the concentration of defects or other thickness related issues.
 - **Response:** Thanks for the reviewer's valuable comments, the thickness of the samples (1.5 mm) have been added in the revised manuscript.

REVIEWER COMMENTS

Reviewer #1 (Remarks to the Author):

The authors reported a giant EC effect near room temperature in a newly developed layered hybrid perovskite ferroelectric [(CH₃)₂CHCH₂NH₃]₂PbCl₄ using an indirect method. But I have several concerns in the process of evaluation of the electrocaloric effect. For indirect method, an overestimation of ΔT is not rare. After carefully read comments from other reviewers, we do agree that there is a growing scepticism to such high temperature change values without validation of direct measurements. My comments are listed below:

1. The authors claimed that they used Maxwell relation for the evaluation of electrocaloric effect of [(CH₃)₂CHCH₂NH₃]₂PbCl₄. However, Maxwell relation can only be used in continuous phase transition (Gujrati et al., Physical Review E 2012, 85, 041129). [(CH₃)₂CHCH₂NH₃]₂PbCl₄ undergoes a first-order phase transition at 302 K which does not fit Maxwell relation (Lu et al., Journal of Inorganic Materials 2014, 29(1), 6-12).

2. The theoretical investigation based on first-principle effective Hamiltonian approach in supporting information claimed that Curie-Weiss fitting are used for the estimation of the first order derivative of dielectric stiffness. However, the component is an improper ferroelectric, the Curie-Weiss law is not applicable in this situation. Wang et al.'s work in the reference used proper ferroelectric [MDABCO](NH₄)I₃, which follows the Curie-Weiss law (Wang et al., Advanced Materials 2020, 32, 1906224). The authors are encouraged to improve the methods which is suitable for improper ferroelectrics.

Over all, since the report of giant EC effect is the most important part in this manuscript and it would cause a very large scale impact on the related research fields, I strongly suggest the author to carry out a direct measurement to evaluate the reported compound.

Reviewer #2 (Remarks to the Author):

The authors have sincerely addressed all of my comments. I recommend the manuscript for publication in Nature Communications.

There are some minor English errors in the manuscript which needs to be addressed prior to publication.

Page 1, Line 13: "high efficient" should be replaced by "highly efficient" or "high efficiency".

Page 2, Line 34: A word is missing between "particularly status"...it could be "particularly important status".

Page 3, Line 47: "emerge" should be replaced by "emerged".

Page 3, Line 60: "had" should be replaced by "has".

Page 6, Line 104: "highly" should be replaced by "high degree of".

Page 6, Line 128 & 261: "derive" should be replaced by "derived".

Page 10, Line 166: "Maxwell relation" should be replaced by "Maxwell's relations".

Page 14, Line 218: The words "purchased used" needs to be corrected.

Page 14, Lines 224 & 226: "were" should be replaced by "was".

Page 15, Line 253: "frequency of 100 K, 300 K, 500 K and 1 MHz" needs to be corrected.

Devajyoti Mukherjee

Reviewer #3 (Remarks to the Author):

The authors have addressed my questions and followed suggestions.
I'm glad to recommend the work for publication in Nature Communications.

Response to Reviewers

Reviewer: 1

Comments:

The authors reported a giant EC effect near room temperature in a newly developed layered hybrid perovskite ferroelectric $[(\text{CH}_3)_2\text{CHCH}_2\text{NH}_3]_2\text{PbCl}_4$ using an indirect method. But I have several concerns in the process of evaluation of the electrocaloric effect. For indirect method, an overestimation of ΔT is not rare. After carefully read comments from other reviewers, we do agree that there is a growing scepticism to such high temperature change values without validation of direct measurements. My comments are listed below:

- **Response:** We deeply thank the reviewer for the time in reviewing the paper and professional feedback. The questions and suggestions are very critical to improve our work. We have tried our best to address the questions and revised our manuscript accordingly.
- (1) The authors claimed that they used Maxwell relation for the evaluation of electrocaloric effect of $[(\text{CH}_3)_2\text{CHCH}_2\text{NH}_3]_2\text{PbCl}_4$. However, Maxwell relation can only be used in continuous phase transition (Gujrati et al., *Physical Review E* 2012, 85, 041129). $[(\text{CH}_3)_2\text{CHCH}_2\text{NH}_3]_2\text{PbCl}_4$ undergoes a first-order phase transition at 302 K which does not fit Maxwell relation (Lu et al., *Journal of Inorganic Materials* 2014, 29(1), 6-12).
- **Response:** Thank you for reviewer's reviewer's valuable comments. Generally speaking, Maxwell's relation is not valid in the case of first-order phase transitions (these are not thermodynamically reversible, *Physical Review E* 2012, 85, 041129). Taking into account the discontinuous change of the polarization at the first order phase transition, EC entropy change ΔS was suggested to be modified as Clausius–Clapeyron relation:

$$\Delta S = \frac{1}{\rho} \int_0^{E_{max}} \left(\frac{\partial P}{\partial T} \right)_E dE + \Delta D \left(\frac{\partial E}{\partial T} \right)$$

From the Equation, the relation can be expressed in two forms: one in which dS is proportional to dE , and the other form in which dS is proportional to the change of the displacement dD having a proportional factor of $\left(\frac{\partial E}{\partial T} \right)$. The equation is an approximate expression, which has not been employed in experimental work due to its technically complex. While, these background effects $\Delta D \left(\frac{\partial E}{\partial T} \right)$ are normally much smaller than caloric effects at transitions of interest, and therefore this method is nominally equivalent to the Maxwell method (*Appl. Phys. Rev.* 2016, 3 (3), 031102; *Nat. Mater.* 2014, 13 (5), 439). As a consequence, The EC effect of $[(\text{CH}_3)_2\text{CHCH}_2\text{NH}_3]_2\text{PbCl}_4$ is evaluated basis on the Maxwell's relations.

- (2) The theoretical investigation based on first-principle effective Hamiltonian approach in supporting information claimed that Curie-Weiss fitting are used for the estimation of

the first order derivative of dielectric stiffness. However, the component is an improper ferroelectric, the Curie-Weiss law is not applicable in this situation. Wang et al.'s work in the reference used proper ferroelectric [MDABCO](NH₄)I₃, which follows the Curie-Weiss law (Wang et al., *Advanced Materials* 2020, 32, 1906224). The authors are encouraged to improve the methods which is suitable for improper ferroelectrics.

- **Response:** Thank you very much for your nice suggestion. In sufficient consideration of the free energy description for improper ferroelectrics, we had modified our computational method for our studied system, (iBA)PbCl₄, based on our CI-NEB calculations in the supporting information to rewrite its free energy expression for phase transition. In our new method, the Curie-Weiss fitting for the first order derivative of dielectric stiffness was not used anymore. Based on the CI-NEB calculations for free energy change and the first-principles calculations for polarization change during phase transition, we employed a specific expansion of free energy to obtain its fitting expression. Finally, this free energy expression is used to recalculate the entropy change and adiabatic temperature change of the improper ferroelectric (iBA)₂PbCl₄. We think this treatment is more reasonable.

Reviewer: 2

Comments:

The authors have sincerely addressed all of my comments. I recommend the manuscript for publication in *Nature Communications*.

There are some minor English errors in the manuscript which needs to be addressed prior to publication.

Page 1, Line 13: "high efficient" should be replaced by "highly efficient" or "high efficiency".

Page 2, Line 34: A word is missing between "particularly status"...it could be "particularly important status".

Page 3, Line 47: "emerge" should be replaced by "emerged".

Page 3, Line 60: "had" should be replaced by "has".

Page 6, Line 104: "highly" should be replaced by "high degree of".

Page 6, Line 128 & 261: "derive" should be replaced by "derived".

Page 10, Line 166: "Maxwell relation" should be replaced by "Maxwell's relations".

Page 14, Line 218: The words "purchased used" needs to be corrected.

Page 14, Lines 224 & 226: "were" should be replaced by "was".

Page 15, Line 253: "frequency of 100 K, 300 K, 500 K and 1 MHz" needs to be corrected.

- **Response:** We deeply thank the reviewer for the time in reviewing the paper and professional feedback. The questions and suggestions are very critical to improve our work. We have revised our manuscript accordingly.

Reviewer: 3

Comments:

The authors have addressed my questions and followed suggestions. I'm glad to recommend the work for publication in Nature Communications.

➤ **Response:** Thank the reviewer for reviewers' kind comments.

REVIEWERS' COMMENTS

Reviewer #1 (Remarks to the Author):

I'm happy to see that the authors have addressed all my concerns and questions. I think this manuscript can be accepted for publication.

Response to Reviewers

Reviewer: 1

Comments:

I'm happy to see that the authors have addressed all my concerns and questions. I think this manuscript can be accepted for publication.

➤ **Response:** We deeply thank the reviewer for the time in reviewing the paper.